# Development of *Spodoptera exigua* Population: Does the Nutritional Status Matter?

**DOI:** 10.3390/insects14010013

**Published:** 2022-12-22

**Authors:** Hancheng Ma, Bin Zhang, Jiangjie Li, Fengjiao Qiao, Qihong Ma, Xuanwu Wan, Zhufeng Jiang, Changyou Li

**Affiliations:** 1Shandong Engineering Research Center for Environment-Friendly Agricultural Pest Management, College of Plant Health & Medicine, Qingdao Agricultural University, Qingdao 266109, China; 2Plant Protection Station of Sichuan Provincial Department of Agriculture and Rural Affairs, Chengdu 610000, China; 3Huashixing Technology Co., Ltd., Qingdao 266109, China

**Keywords:** nutrition, population life table, *Spodoptera exigua*, protein, carbohydrates

## Abstract

**Simple Summary:**

*Spodoptera exigua* (Hübner) is a common agricultural pest that can harm hundreds of crops. Carbohydrates and proteins are the two key nutrients affecting the development of insect populations. To study the development of the *S. exigua* population under different carbohydrate and protein nutrition conditions, we constructed a life table of *S. exigua* under different nutrient conditions: high nitrogen, medium nitrogen–medium sugar, and high sugar. The results showed that there was no significant difference in net reproductive rate among the three nutritional conditions. *S. exigua* on medium nitrogen–medium sugar diets had the shortest generation time and significantly higher intrinsic rate of increase, finite rate of increase, fecundity, and predicted population at 100 days. These results suggested that an appropriate protein:carbohydrate ratio was beneficial to the rapid development of *Spodoptera exigua* on farmland. These findings are important for scientifically predicting the population dynamics of *S. exigua* and constructing a system for controlling the insect population.

**Abstract:**

*Spodoptera exigua* (Hübner) is a common agricultural pest that can harm hundreds of crops. Nutrition conditions can significantly affect the development of insects, especially carbohydrates (C) and proteins (P), which are the two most critical nutrients. To study the development of the *S. exigua* population under different carbohydrate and protein nutrition conditions, we constructed a life table of both sexes of an *S. exigua* population under three artificial diets: high nitrogen (P:C = 7:1), medium nitrogen, medium sugar (P:C = 1:1), and high sugar (P:C = 1:7). The results showed that the generation time of *S. exigua* was 26.38 ± 0.54 d under the medium nitrogen–medium sugar diet, which was the shortest among the three nutrition conditions. The intrinsic rate of increase (0.18 ± 0.01), finite rate of increase (1.20 ± 0.01), fecundity (605.42 ± 36.33 eggs/female), and predicted population at 100 days (8,840,000) were significantly higher under the medium nitrogen–medium sugar condition. There was no significant difference in the net reproductive rate among the three conditions. These results suggested that an appropriate protein:carbohydrate ratio is beneficial to the rapid development of *S. exigua* on farmland. These findings are important for scientifically predicting the population dynamics of *S. exigua* from the perspective of nutritional ecology, understanding its catastrophic mechanism, and constructing a prevention and control system.

## 1. Introduction

In nature, nutrition is one of the main environmental factors regulating the growth and development of insects [1]. Different nutrition conditions can shape various patterns in ecology and evolution. Nutrient fluctuations can directly affect the rate of insect growth and development, and even affect the probability of diapause, which can be induced by low-quality nutrients [2]. Inter- and intraspecific variation in nutrition can impact herbivore fecundity [3], and shape the generalist–specialist dynamics [4,5,6]. Nutrition can also affect insect life span and population [7,8,9].

Protein (P) and carbohydrates (C) are two major essential nutrients, and the content and ratio of these two nutrients have important effects on insect growth and population development [10]. Carbohydrates provide energy through glycolysis, store energy in the form of glycogen [11], and provide raw materials and precursors for chitin synthesis and lipid conversion [12]. Protein is an essential substance for insect growth and development and influences insect pupa weight and adult egg laying [13], but the amounts and ratios of these two macronutrients in plants are variable and affected by many factors, including plant species, growth periods, and environmental conditions (water, fertilizer, light, etc.). Even different parts of the same plant have different contents and proportions of nutrients [14,15]. To maintain the survival and development of the population, insects adopt different foraging strategies to obtain optimal nutrition under different environmental conditions, which helps to meet the needs of optimal population growth. Experimental evidence collected over the past several decades has shown that insects are able to self-select a mixture of nutrients that supports optimal performance [16,17,18]. During the evolutionary process, insects have developed a nutritional regulation mechanism suitable for the population development. Under optimal nutritional conditions, individual insects exhibit faster growth rate and lower mortality. Based on previous research, Raubenheimer and Simpson (2018) summarized four theoretical models used to describe insect response strategies to nutrient stress, and named it the geometric framework (GF) [19]. With the GF model, researchers have elucidated the preferred carbohydrate to protein ratios (hereinafter referred to as P:C) of many insects [19,20].

The beet armyworm, *Spodoptera exigua* (Hübner), belongs to the Lepidoptera of Noctuidae. It is an important agricultural pest and can harm many plants, including several major crops such as sugar beet, cotton, soybean, and potatoes [19,21,22]. The young larvae always feed on terminal clusters, seedlings, and stems of host crops [23]. Serious outbreaks of this pest in many regions of Asia, Africa, Europe, and the US have been reported in the past two decades [24]. This insect has a wide range of host plants, high fecundity, and migratory nature; it is also developing resistance to pesticides [24,25]. Thus, a new method is needed to control the beet armyworm. Studies have confirmed that the amount of nitrogen fertilizer applied to farmland changes the nutrient structure of farmland ecosystems by directly affecting carbon-nitrogen ratios and levels in plants. This in turn significantly affects the growth and population dynamics of herbivorous insects, including the beet armyworm [12,26]. Systematic study on the nutritional ecology of *S. exigua* is helpful for in-depth understanding of its catastrophic mechanism and constructing a prevention and control system. At present, most of the studies on insect nutritional ecology use artificial diets as a nutrient source [27]. This method can not only eliminate the stress of plant allelochemicals and easily quantify the stress effect of nutrition but also allow research using artificial diets and field applications to support each other [28]. Therefore, it is necessary and beneficial to use artificial diets to conduct research on insect nutrition ecology.

In this study, we used *S. exigua* as the research object, and constructed the age-stage specific life table of *S. exigua* under artificial diets with different carbon to nitrogen ratios. The results not only help to deeply understand how the nutrient structure of the farmland ecosystem affects *S. exigua* catastrophe and improve the theoretical system of insect nutrition ecology but also help to construct a scientific system for controlling polyphagous noctuid pests such as *S. exigua*.

## 2. Materials and Methods

### 2.1. Collecting and Rearing of Test Insects

*S. exigua* was collected from a green onion field in Shangma, Chengyang District (Qingdao, Shandong, China) (36.27° N, 120.17° E). On the basis of the artificial diet formula of *S. exigua* [29], we changed the contents of yeast powder and glucose to set three nutritional treatments with different protein (P) to carbohydrate (C) ratios: high nitrogen (P:C = 7:1), medium nitrogen, medium sugar (P:C = 1:1), and high sugar (P:C = 1:7). In these three nutritional treatments, we continuously reared multiple generations of insects. The artificial rearing conditions were 28 °C ± 1 °C, 60% ± 5% humidity, and L14:D10 photoperiod.

### 2.2. Experimental Method

After the adult insects laid eggs and hatched under the three nutritional conditions, 60 newly hatched larvae were taken from each condition and reared in a circular small box (bottom area × height: 7 mm^2^ × 3 mm = 21 mm^3^). Fresh artificial diet was replaced every day, and the survival and development of larvae were observed and recorded until pupation stage. The development and survival of pupae were recorded after pupation until eclosion. After eclosion, females and males were put into a square box for mating and fed with 10% honey water. The honey water was changed every day, and each box was guaranteed to have 1 pair of adults. If the male died, the male insects of the same age would be supplemented from the corresponding nutrient population, and the death time would not be recorded; if the female died, no supplementation was required, and the time of death was recorded. The number of eggs laid and the survival of adults were observed and recorded daily.

### 2.3. Statistical Analysis of Data

The population life table analysis was carried out according to the bisexual population life table analysis method [30,31,32]. Specifically, TWOSEX-MSChart software was used to analyze the raw data [33,34]. The spawning period, average pre-oviposition period (APOP), total pre-oviposition period (TPOP), fecundity, intrinsic rate of increase (*r*), finite rate of increase (*λ*), net reproductive rate (*R*_0_), average generation period (*T*), and other population life table parameters were calculated, and the differences between nutritional conditions were analyzed (*p* < 0.05). The age-stage survival rates (*s_xj_*, *x* = age, *j* = stage), age-stage fecundity (*f_xj_*), age-specific survival rate (*l_x_*), age-specific fecundity (*m_x_*), age-specific life expectancy (*e_xj_*), age-stage reproductive value (*v_xj_*) and other specific developmental stage dynamic parameters were calculated. For all the above parameters, bootstrap = 100,000 method can be used to evaluate the standard error of the parameter. SigmaPlot 12.0 (Systat Software, Inc., San Jose, CA, USA) software was used for graphical display of the analysis results.

## 3. Results

### 3.1. Developmental Duration of S. exigua under Different Nutritional Conditions

The bisexual life table analysis (Table 1) showed that the duration of first-instar larvae under high sugar was significantly longer than that under high; the duration of second-instar larvae under middle nitrogen, middle sugar was significantly shorter than that under high sugar and high nitrogen; the duration of third-instar larvae under high nitrogen was significantly longer than the other two conditions; the duration of fourth-instar larvae under medium sugar, medium nitrogen was significantly shorter than that under high nitrogen; the duration of fifth-instar larvae and pupal stage was not significantly different between the three treatments; the adult life span under medium nitrogen, medium sugar (8.78 d) was significantly longer than that under high nitrogen; and the preadult stage (22.04 d) under medium nitrogen, medium sugar was significantly shorter than that under high nitrogen.

### 3.2. The Fecundity Parameters of S. exigua under Different Nutritional Conditions

The bisexual life table analysis showed that there were significant differences in the fecundity parameters of *S. exigua* under the three nutritional conditions. The oviposition days was not obviously different among the three nutrient conditions. The total pre-oviposition period (TPOP) was 24.38 d under medium nitrogen, medium sugar, which was significantly shorter than that under high sugar (26.75 d) and high nitrogen (27.55 d). The average pre-oviposition period (APOP) under high nitrogen (2.36 d) was significantly shorter than that under medium nitrogen, medium condition (3.23 d) and high sugar (3.88 d). The fecundity (egg production per female) under medium nitrogen, medium sugar (605.42 eggs/female) was significantly higher than that under high sugar (358.90 eggs/female) and was not significantly different from high nitrogen (486.89 eggs/female) (Table 2).

### 3.3. Life Table Parameters of S. exigua Population under Different Nutritional Conditions

The intrinsic rate of increase (*r*) is an important parameter to measure the population growth, which can comprehensively reflect the increase, reproduction, and survival of the population [35]. Our results (Table 3) showed that the intrinsic rate of increase under medium nitrogen, medium sugar (0.18) was significantly higher than that under high sugar (0.15) and high nitrogen (0.15). The finite rate of increase, net reproductive rate, and mean generation time were also important factors for population growth. We found that the finite rate of increase (*λ*) was significantly higher under medium nitrogen, medium sugar (1.20) than that under high sugar (1.16) and high nitrogen (1.16); the mean generation time was significantly shorter under medium nitrogen, medium sugar (26.38) than that under high sugar (28.30) and high nitrogen (29.50) (Table 3); the net reproductive rate (*R*_0_) was not significantly different between the three nutrient conditions.

### 3.4. Age-Stage Survival Rate and Fecundity of S. exigua under Different Nutritional Conditions

We found that the survival rates overlapped between developmental stages, indicating that the individual development of each population was not uniform. The age-stage survival rate (*s_xj_*) represents the survival status at age *x* stage *j*. As shown in Figure 1, under high nitrogen, the peak survival rate was lower at the fourth, fifth and pupal stages, and the adult eclosion time was relatively late.

The age-specific survival rate *l_x_* represents the survival state of all individuals, ignoring the difference in stage; *m_x_* is the age-specific fecundity, reflecting the average number of laid eggs of the entire population. Therefore, *l_x_m_x_* represents the net maternity of the entire population. The results showed that the net maternity was significantly higher under medium nitrogen, medium sugar (44.70) than that under high sugar (22.03) and high nitrogen (30.25) (Figure 2).

### 3.5. The Age-Specific Life Expectancy of S. exigua Population under Different Nutritional Conditions

The life expectancy (*e_xj_*) represents the expected future life span of an individual at age *x* stage *j*. Life expectancy is used to predict the survival time of individuals in the future, which is important for evaluating the degree of damage to individuals. Life expectancy gradually decreases as the population ages and eventually approaches zero. Our results showed that eggs, first-instar larvae, and second-instar larvae had the highest life expectancy under high sugar (23.38 d, 21.38 d and 20.06 d), while the third-instar larvae, fifth-instar larvae, and pupae had the highest life expectancy under medium nitrogen, medium sugar (20.68 d, 17.88 d and 14.55 d). The life expectancy of the fourth-instar larvae was the highest under high nitrogen (18.82 d), and the life expectancies of female and male larvae were the highest under high nitrogen and medium sugar, medium nitrogen, respectively (10.91 d and 9.30 d).The eggs, first-instar larvae, second-instar larvae, third-instar larvae, fifth-instar larvae, and males had the lowest life expectancies under high nitrogen (19.68 d, 17.68 d, 18.27 d, 17.73 d, 16.72 d and 7.57 d), while the fourth-instar larvae, pupae, and females had the lowest life expectancies under high sugar (18.55 d, 13.16 d and 9.64 d) (Figure 3).

### 3.6. The Reproductive Value of S. exigua under Different Nutritional Conditions

The reproductive value (*v_xj_*) describes the contribution of an individual at age *x* stage *j* to future population growth. As shown in Figure 4, the highest reproductive value (369.68 d^−1^) under medium nitrogen, medium sugar occurred on the 24th day; under high sugar, the highest reproductive value (270.86 d^−1^) appeared on the 26th day; and under high nitrogen, the highest reproductive value (385.55 d^−1^) appeared on the 27th day.

### 3.7. The Prediction of S. exigua Population Size under Different Nutritional Conditions

We simulated the 100-day population size of *S. exigua* and found that the population grew the fastest under medium nitrogen, medium sugar. At 100 days, the *S. exigua* population reached 8.84 million under medium, nitrogen medium sugar, 2.58 million under high sugar, and 4.8 million under high nitrogen (Figure 5). This result further demonstrated that medium nitrogen, medium sugar was more beneficial to population growth, and a balanced carbon to nitrogen ratio could better meet the requirements for rapid generation change.

## 4. Discussion

The results of the developmental duration of each stage showed that under medium sugar, medium nitrogen, the preadult stage of *S. exigua* was shorter, indicating that during most of the larvae developmental stages, appropriate ratio of protein and sugar can facilitate rapid population development of *S. exigua*. The results also showed that under medium sugar, medium nitrogen, oviposition days were not significantly different and the APOP was not the shortest among the three conditions; however, the TPOP and egg laying peak were the earliest and the net maternity was higher than the other two conditions. These findings indicated that medium sugar, medium nitrogen helped the population to quickly complete reproduction and produce more offspring to occupy an ecological niche. These findings are consistent with previous research results that the optimal carbon-nitrogen ratio of *S. exigua* is 1:1.1, as obtained by the GF method [36].

Insects are forced to (especially in agroecosystems) cope with nutritional stress caused by nutrient deficiencies quite frequently. Therefore, they have to develop many strategies to deal with the nutritional stress. The strategies, including intake preference and postintake nutrient regulation, aim to help insects obtain optimal nutrition, find better growth situations, and gain stronger adaptation to environmental conditions [37]. For example, some polyphagous Orthoptera insects prefer foods with high sugar content to ensure sufficient energy for more activities [20,28], while some polyphagous lepidopteran larvae prefer foods with high protein content [28,36,38].

In addition, in a previous study, we found that young *S. exigua* larvae tended to feed on high-protein diets, while older larvae preferred high-sugar diets [39]. This is consistent with the feeding characteristics of many lepidoptera in the field, that is, young larvae prefer young leaves with higher protein content, and older larvae prefer mature leaves with higher sugar content [40]. Meanwhile, the feeding preference can alter with the environmental stress. In our previous study, when *S. exigua* was infected with nucleopolyhedrovirus, they significantly changed their feeding preference, in which both younger and older larvae tended to feed on diets with a high P:C ratio [41].

In this study, we confirmed that the P or C biased diet can be regarded as nutrition stress for *S. exigua* larva. In subsequent study, we will focus on how *S. exigua* regulates the nutrition preference when it suffers from nutrition stress. Studies have found that polyphagous Lepidoptera insects have stronger postfeeding nutrient regulation than oligophagous insects [37]. For example, a study on *Manduca sexta* found that the blood sugar level in the insect was closely related to feeding preference and could be regulated through the gluconeogenesis pathway [42]. Understanding the metabolic regulation of *S. exigua* under nutritional stress is important for discovering new molecular targets for pest control. It also allows us to adjust the nutrient structure in the field through feeding preferences and change the population development of *S. exigua*. Moreover, it helps us to understand the mechanism of insect outbreaks from a nutritional ecology point of view.

## 5. Conclusions

This study investigated the population development of the *S. exigua* under different nutrient conditions. The results clearly demonstrated the importance of nutrition conditions for *S. exigua* population growth. Under the condition of medium, nitrogen medium sugar, the mean generation time of *S. exigua* was the shortest, and the intrinsic rate of increase, finite rate of increase, net maternity, and predicted population at 100 d were the highest. These findings not only have important implications for understanding the outbreak mechanisms of *S. exigua* from a nutritional ecology perspective but also help in discovering new molecular targets for pest control based on nutrient utilization pathways.

## Figures and Tables

**Figure 1 insects-14-00013-f001:**
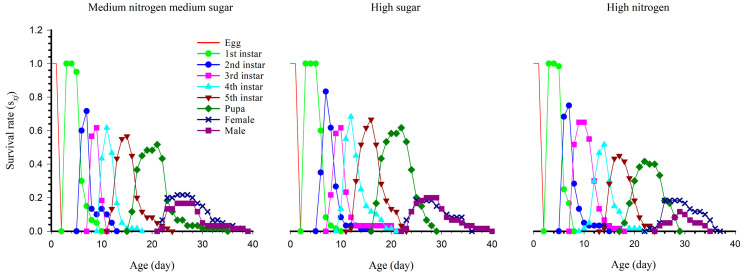
The age-stage survival rate (*s_xj_*) of *S. exigua* under three nutritional conditions.

**Figure 2 insects-14-00013-f002:**
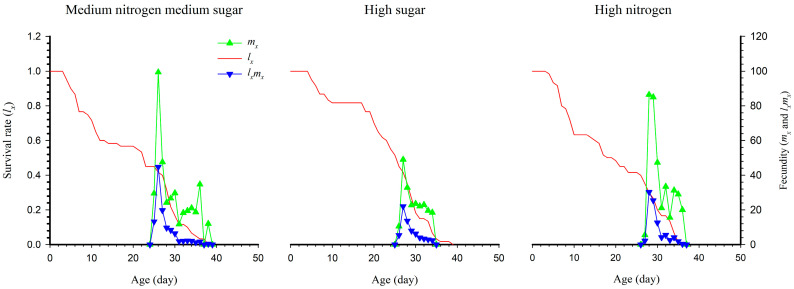
The age-specific survival rate (*l_x_*), fecundity (*m_x_*) and net maternity (*l_x_m_x_*) of *S. exigua* under three nutritional conditions.

**Figure 3 insects-14-00013-f003:**
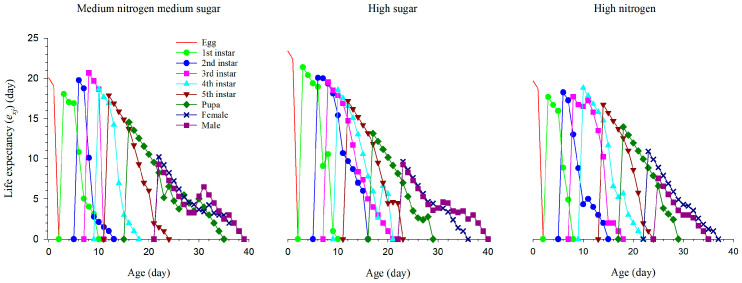
The age-special life expectancy (*e_xj_*) of *S. exigua* under three nutritional conditions.

**Figure 4 insects-14-00013-f004:**
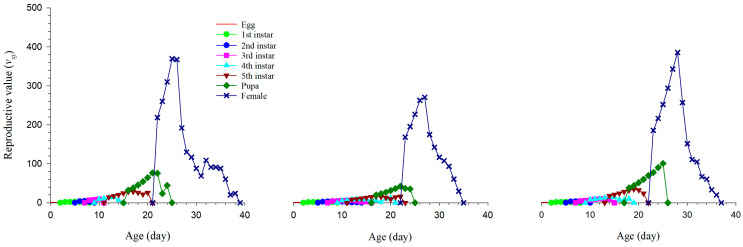
The reproductive value of *S. exigua* (*v_xj_*) under three nutritional conditions.

**Figure 5 insects-14-00013-f005:**
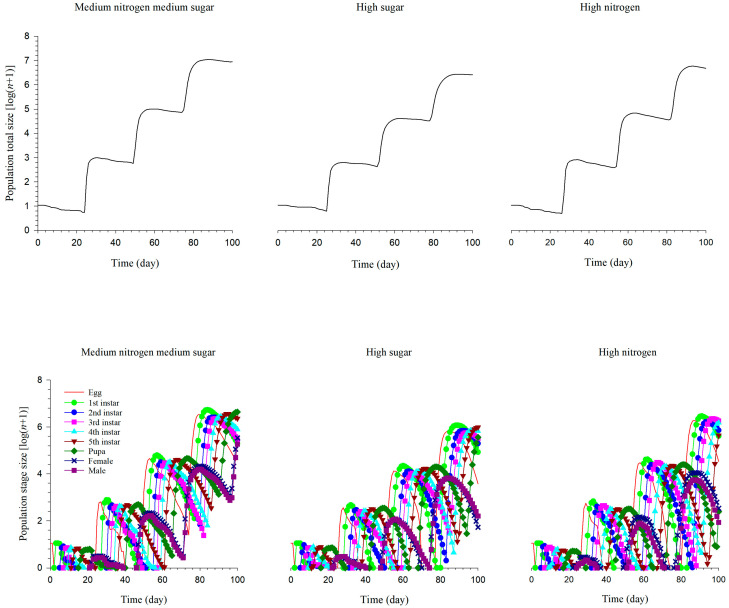
The prediction of *S. exigua* population size under different nutritional conditions.

**Table 1 insects-14-00013-t001:** Developmental duration of *S. exigua* under three nutritional conditions.

Developmental Stage (d)	Medium Nitrogen, Medium Sugar	High Sugar	High Nitrogen
First-instar larva	3.53 ± 0.16 ^a,b^	3.66 ± 0.08 ^a^	3.21 ± 0.08 ^b^
Second-instar larva	1.97 ± 0.03 ^b^	2.65 ± 0.16 ^a^	2.40 ± 0.16 ^a^
Third-instar larva	2.22 ± 0.08 ^b^	2.23 ± 0.10 ^b^	4.34 ± 0.17 ^a^
Fourth-instar larva	2.79 ± 0.08 ^b^	3.00 ± 0.14 ^a,b^	3.17 ± 0.14 ^a^
Fifth-instar larva	4.39 ± 0.15 ^a^	4.63 ± 0.17 ^a^	4.48 ± 0.17 ^a^
Pupa	6.13 ± 0.14 ^a^	6.50 ± 0.15 ^a^	6.22 ± 0.19 ^a^
Adult lifespan	8.78 ± 0.63 ^a^	7.96 ± 0.63 ^a,b^	7.22 ± 0.70 ^b^
Preadult stage	22.04 ± 0.18 ^b^	23.50 ± 0.28 ^a,b^	25.17 ± 0.35 ^a^

^a,b^ Different lowercase letters indicate significant differences between different nutritional conditions for the same parameter (*p* < 0.05).

**Table 2 insects-14-00013-t002:** Fecundity parameters of *S. exigua* under three nutritional conditions.

Parameters	Medium Nitrogen, Medium Sugar	High Sugar	High Nitrogen
Oviposition days (d)	6.00 ± 0.55 ^a^	4.75 ± 0.56 ^a^	5.19 ± 0.64 ^a^
TPOP (d)	24.38 ± 0.35 ^b^	26.75 ± 0.53 ^a^	27.55 ± 0.31 ^a^
APOP (d)	3.23 ± 0.20 ^a^	3.88 ± 0.61 ^a^	2.36 ± 0.15 ^b^
Fecundity (eggs/female)	605.42 ± 36.33 ^a^	358.90 ± 94.50 ^b^	486.89 ± 64.82 ^a,b^

^a,b^ Different lowercase letters indicate significant differences between different nutritional conditions for the same parameter (*p* < 0.05).

**Table 3 insects-14-00013-t003:** Life table parameters of *S. exigua* population under three nutritional conditions.

Parameters	Medium Nitrogen, Medium Sugar	High Sugar	High Nitrogen
Intrinsic rate of increase (*r*)	0.18 ± 0.01 ^a^	0.15 ± 0.02 ^b^	0.15 ± 0.01 ^b^
Finite rate of increase (*λ*)	1.20 ± 0.01 ^a^	1.16 ± 0.02 ^b^	1.16 ± 0.01 ^b^
Net reproductive rate (*R*_0_)	3.79 ± 0.24 ^a^	4.83 ± 1.35 ^a^	4.62 ± 1.89 ^a^
Mean generation time	26.38 ± 0.54 ^b^	28.30 ± 0.62 ^a^	29.50 ± 0.26 ^a^

^a,b^ Different lowercase letters indicate significant differences between different nutritional conditions for the same parameter (*p* < 0.05).

## Data Availability

The data presented in this study are available on request from the corresponding author.

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
