# Peer review of "Development of Spodoptera exigua Population: Does the Nutritional Status Matter?"

_insects, 2022, doi:10.3390/insects14010013_

Round 1
Reviewer 1 Report
Reviewer’s comments:
Page 2, line 103
Printed: “...artificial diet formula of S. exigua (Xiao et al., 2005), we changed the contents ...”
The paper of Xiao et al. (2005) is absent in References list, but it is very important for methods. Thus, replace (Xiao et al., 2005) for [29], change all numbers of cited paper after line 124 of manuscript from [29-43] to [30-44] respectively.
In References list (Pages 10-11) add paper No 29:
29. Xiao et al., 2005 .....
and change numbers of papers from [29-43] to [30-44] respectively.
In Conclusions (Page 9, lines 293-294) seems to be better replace “The results showed that the importance of nutrition conditions for S. exigua population.” for “The results showed the importance of nutrition conditions for S. exigua population.”
Author Response
"Please see the attachment."

Reviewer 2 Report
Thanks to the authors for an interesting manuscript. My comments and recommendations are in the attached file.
The work is interesting from the point of view of practical entomology. It expands the understanding of insect nutrition and can serve as a basis for the development of pest control programs and mechanisms.
As a recommendation, I would like to ask the authors to attach several photographs with different stages of development on different types of diets.

Author Response
"Please see the attachment."

Reviewer 3 Report
The authors investigated the effect of artificially controlled feeding on the population growth of Spodoptera exigua. In general the manuscript reads well and the results are well presented. However, the results are not well set into context and compared with each other. Most importantly, the authors do not set the choice of artificial diets into a natural comparison. What is the control? How do the diets compare with natural uptake of nitrogen and proteins?
Title: the title is not specific. "Performance of...population" could mean anything. The manuscript tested population growth.
Methods: please provide references to the software used (ie. Company, Brand location).
Discussion: It is not clearly presented how the different nutritional intakes compare with a) natural intakes and b) with each other. Is 1:1 mixture of P:C more natural than the strongly derived ones? The findings are said to be of importance to pest management. Without a comparison to the actual nutrition situation it is hard to guess which implications those might be. We learn that a medium combination leads to significantly higher growth, but the result primarily suggests that we can improve population growth by setting a balanced P:C intake, which seems not to be the original question of the study. It is necessary to discuss the findings more reflected and weight the benefits and differences for population growth among the diets and how these are informative for field based operations with pest insects.
Author Response
"Please see the attachment."

Round 2
Reviewer 3 Report
The first comments were solved more or less satisfactorily.
The discussion however, still needs some improvement: as confirmed in the authors response the actual finding is, that S. exigua populations develop better in a balanced P:C food composition, but less in sugar or nitrogen biased intakes. This finding is more or less emphasized in the discussion, but needs to be sharpened to set these findings into a context. The development in the two other feeding compositions was less efficient as with balanced ratios, but still not drastically reduced. What can be drawn from this finding?
Furthermore I encourage to highlight potential conclusions that would be useful for pest management. Please offer cosnequences for agriculture, that might be possible to be drawn from the findings to manage S. exigua. Otherwise the data would only provide the statement, that, if you would like to mass rear S. exigua, offer balanced food.
The discussion should be properly proofred, there are still many minor expression issues.
Author Response
请参阅附件
